# Fan-Out Wafer and Panel Level Packaging as Packaging Platform for Heterogeneous Integration

**DOI:** 10.3390/mi10050342

**Published:** 2019-05-23

**Authors:** Tanja Braun, Karl-Friedrich Becker, Ole Hoelck, Steve Voges, Ruben Kahle, Marc Dreissigacker, Martin Schneider-Ramelow

**Affiliations:** 1Fraunhofer Institute for Reliability and Microintegration, 13355 Berlin, Germany; Karl-Friedrich.Becker@izm.fraunhofer.de (K.-F.B.); Ole.Hoelck@izm.fraunhofer.de (O.H.); Steve.Voges@izm.fraunhofer.de (S.V.); Ruben.Kahle@izm.fraunhofer.de (R.K.); 2Forschungsschwerpunkt Technologien der Mikroperipherik, Technical University Berlin, 13355 Berlin, Germany; marc.dreissigacker@izm.fraunhofer.de (M.D.); Martin.Schneider-Ramelow@izm.fraunhofer.de (M.S.-R.)

**Keywords:** fan-out wafer level packaging, panel level packaging, heterogeneous integration

## Abstract

Fan-out wafer level packaging (FOWLP) is one of the latest packaging trends in microelectronics. Besides technology developments towards heterogeneous integration, including multiple die packaging, passive component integration in packages and redistribution layers or package-on-package approaches, larger substrate formats are also targeted. Manufacturing is currently done on a wafer level of up to 12”/300 mm and 330 mm respectively. For a higher productivity and, consequently, lower costs, larger form factors are introduced. Instead of following the wafer level roadmaps to 450 mm, panel level packaging (PLP) might be the next big step. Both technology approaches offer a lot of opportunities as high miniaturization and are well suited for heterogeneous integration. Hence, FOWLP and PLP are well suited for the packaging of a highly miniaturized energy harvester system consisting of a piezo-based harvester, a power management unit and a supercapacitor for energy storage. In this study, the FOWLP and PLP approaches have been chosen for an application-specific integrated circuit (ASIC) package development with integrated SMD (surface mount device) capacitors. The process developments and the successful overall proof of concept for the packaging approach have been done on a 200 mm wafer size. In a second step, the technology was scaled up to a 457 × 305 mm^2^ panel size using the same materials, equipment and process flow, demonstrating the low cost and large area capabilities of the approach.

## 1. Introduction

Within the European funded project smart-MEMPHIS, the goal was to tackle the main challenge for all smart devices—becoming self-powering. The project was aimed to design, manufacture and test a miniaturized autonomous energy supply based on harvesting vibrational energy with piezo-MEMS energy harvesters. Cost effective packaging was needed for the 3D system integration of a MEMS-based multi-axis energy harvester, an ultra-low-power ASIC (application-specific integrated circuit) to manage the variations of the frequency and harvested power, and a miniaturized energy storing supercapacitor [1,2]. Miniaturization was another key demand as target applications were a leadless pacemaker and a wireless sensor network for structural health monitoring. Panel level packaging (PLP) was selected as the packaging technology for the single components. The ASIC, together with two capacitors, have been integrated by a fan-out panel level packaging (FOPLP) approach and will be described in detail in this paper. 

The drivers for 3D packaging solutions are manifold, and each requirement calls for different answers and technologies. The main goal is miniaturization, but the increased component density and performance, simplification of design and assembly, flexibility, functionality and, finally, cost and time-to-market, have been found to be the core drivers for going 3D as well. Besides die and package stacking and folded packages, embedding dies is a key technology for heterogeneous system integration [3].

There are two main approaches for embedded die technologies: fan-out wafer level packaging (FOWLP), where dies are embedded into polymer encapsulants, and PCB (printed circuit board) embedding, where dies are embedded into printed circuit boards [4,5]. A lot of activities that are running worldwide deal with fan-out wafer level integration. Main approaches here include the embedded wafer level ball grid array (eWLB) by Infineon [6], the InFO package by TSMC [7] or the redistributed chip package (RCP) by Freescale [8]. Fan-out wafer level packaging (FOWLP) has been initiated in volume production for mobile and wireless applications (mainly wireless basebands) and is now moving into automotive and medical applications.

One driver for panel level packaging is now, of course, the lowering of costs by increasing the packaging size from wafers to larger panel formats and thereby increasing the number of packages manufactured in parallel. Additionally, PLP has the opportunity to adapt processes, materials and equipment from other technology areas. Printed circuit board (PCB), liquid crystal display (LCD) or solar equipment are manufactured on panel sizes and also offer new approaches for fan-out panel level packaging (FOPLP) [9,10,11].

Figure 1 shows an overview of the typical panel sizes used in PCB and LCD manufacturing in comparison to standard wafer sizes. This already indicates the variety of possible formats without taking the technical challenges and possible limitations for FOPLP into account.

## 2. Process Considerations

For fan-out wafer and panel level packaging, two basic process flows are encountered: the “Mold first” and the “RDL (redistribution layer) first” approaches. By now, for the “Mold first” process, a face-down and a face-up option exist. Both variants are already in mass production. The process flows for all of the options are summarized in Figure 2.

“Mold first” face-down starts with a die assembly on an intermediate carrier, followed by the over-molding and debonding of the molded wafer/panel from the carrier. The redistribution layer, typically based on thin film technology, is finally applied on the reconfigured molded wafer/panel. The face-up approach also starts with a die assembly on a carrier with a temporary adhesive layer. However, for this approach the dies have a Cu-bump and are placed face-up on the carrier. After over-molding, a back-grinding step opens access to the Cu-bumps of the dies again. The redistribution is applied and finally the wafer is released from the carrier and diced for package singulation.

The “RDL first” process is comparable to an advanced flip chip on flex assembly. Here, the redistribution layer is first of all applied on an intermediate carrier, and the bumped dies are assembled by chip to wafer bonding on the RDL. Afterwards, the assembly is underfilled and over-molded and the molded wafer, including RDL, has to be released from the carrier.

Besides the different processes having pros and cons, such as the costs, yield and flexibility, the final package structures also show differences (see Figure 3).

The “mold first” face-down approach has the shortest interconnect with a direct plated via. This may lead to the best RF performance at higher frequencies due to there being lowest loss, especially when the chip to chip connection is considered. “Mold first” face-up needs a Cu pillar in combination with a plated via, and “RDL first” even needs a soldered interconnect. Additionally, the last two options also have an additional polymer/underfill layer between the die and RDL, which may influence the performance and reliability.

For all the different fan-out approaches, there are activities running worldwide on panel sizes. Several companies, including NEPES, POWERTECH, SAMSUNG Electro-Mechanics, DECA Technologies or ASE, have already announced their work on processes ready for high volume manufacturing [12,13,14,15].

## 3. Materials and Methods

For the ASIC packaging, a “mold first” face-down approach was selected, as no additional chip preparation such as bumping was needed and the direct integration of the SMD capacitors is feasible. 

For the package type, a land grid array (LGA) was chosen for a minimum package stand-off in the final system integration. For the process development, dies with same size and IO (input/output) pattern as the functional ASICs were used. The package design with these dies was done to allow the simply daisy chain testing of the interconnects. A second design was done to allow a full functional ASIC packaging. The ASIC die size was 2.01 × 2.01 mm^2^, and the final package size was 2.25 × 1.5 mm^2^. Two off-the-shelf capacitors from muRata (GCG155R71C104KA01, Murata Manufacturing Company, Ltd., Kyoto, Japan) were also integrated into the package. In Figure 4, the daisy chain, as well as the full functional package designs, are shown.

The process developments and the overall proof of concept for the ASIC fan-out wafer/panel level packaging approach was done on a 200 mm wafer size. All materials, equipment and processes were selected and evaluated for direct upscaling to a 457 × 305 mm^2^ panel size. 

### 3.1. Assembly

The carriers were prepared in a first step with a thermal release tape by lamination. For the die placement, an ASM (ASM Assembly Systems GmbH & Co. KG) Siplace CA3 was used that allows a die placement on a wafer but also on a panel in one step. ASICS were directly picked from the diced wafer and placed face down on the carrier. The SMD components were assembled in the same step. On the wafer level, a 100% area utilization was applied, whereas on the panel level the area utilization was only 25% due to the limited number of available dies. Figure 5 depicts examples of the wafer and panel level assembly.

### 3.2. Compression Molding

For the reconfigured wafer encapsulation, compression molding is mainly used [16]. Recent machine developments now also allow panel molding for sizes in the range of 600 × 600 mm^2^. A compression molding evaluation within this study was performed on 200 mm with a wafer level machine from TOWA (TOWA Corporation, Kyoto, Japan) and with a large area panel mold machine from APIC Yamada (APIC YAMADA CORPORATION, Nagano, Japan), using a tooling with a cavity size of 457 × 305 mm^2^.

There are a variety of epoxy molding compounds (EMC) for embedded wafer and panel level molding from different suppliers available on the market. Basically, state of the art materials can be divided into liquid, granular and sheet compounds. In order to find suitable epoxy molding compounds (EMC), together with optimized machine parameters for compression molding and die embedding, mold flow simulations were performed. Sub-models were used to investigate the forces that dies were subjected to during the encapsulation process to avoid significant die shifting of flying dies during the compression [17].

The complex interplay of the shear-thinning behavior and the crosslinking kinetics of the highly filled epoxy resins, with an increasing viscosity, are modeled by incorporating fitted experimental data from plate-plate rheometry and DSC (differential scanning calorimetry) measurements. The procedure for finding the set of optimized parameters will be described in the following.

These models allow us to simulate the compression molding process and yield pressure, as well as the shear-stress on the component’s surfaces. For a die where the front (and back) are perpendicular to the melt front, the resulting force only has pressure contributions, while the sides only have shear contributions. This leads to the following expression:(1)$$F=\left(p_{front}-p_{back}\right)A_{front}+\tau_{top}A_{top}+2\tau_{side}A_{side}$$

In order to keep the stress on the components as low as possible, different combinations of materials, process temperatures and compression profiles were simulated. Through a careful choice of the compression profile, together with a suitable temperature and an EMC with a low viscosity and strong shear-thinning properties, a set of optimized parameters was determined. Figure 6 shows the stress on components during encapsulation at 125 °C for 7 different EMCs. It becomes clear that the materials represented by the yellow and blue graphs are less suitable than the low-viscosity materials which can be seen below them.

Based on this study, we selected a liquid EMC with a resulting low stress on dies and SMDs during the compression molding. The material data are summarized in Table 1.

After the compression molding and post mold cure of the EMC, the reconfigured wafer and panels have been de-bonded from the carrier by a temperature step.

### 3.3. Automated Layout Adaptation

Die shifting is one of the key challenges during “mold first” FOWLP and PLP. Due to the different thermo-mechanical properties of the carrier, the thermo-release tape and epoxy molding compound dies move such that the die position is shifted with respect to the placement position after cooling down from the compression molding. This effect is also influenced by the chemical shrinkage of the molding compound.

Die shifting can be overcome by correcting the initial placement position according to a measured shifting factor, so that after the molding the dies have the correct layout position. However, this approach implies a cost and time intensive process set-up.

Another approach could be to use a fast AOI (automated optical inspection) in combination with maskless processing for die connection and rewiring. This would give the opportunity to tolerate a larger die misplacement by adapting the layout to the real die position. The layouts of the proposed processes, through the mold and blind via, by laser drilling and redistribution structuring by laser direct imaging (LDI), can be automatically adapted according to the measured die positions. Although these process steps are mask-less, they can be highly productive. 

The second approach has been selected for this project. For all of the processed samples, the exact die position has been measured using a Mahr (Mahr GmbH, Göttingen, Germany) OMS 600. A software routine has been used to automatically adapt the layout of the redistribution layer. The layout adaption has been done in in x- and y-direction, but the rotation of the dies were also taken into account.

### 3.4. Redistribution Layer

The redistribution layer (RDL) is based on thin film technology. A photosensitive, low temperature cure polyimide dielectric dry film is laminated and structured by laser direct imaging (LDI). On the one hand, dry film allows for the lamination and application on the reconfigured wafer through the mold vias; on the other hand, the process is directly transferable to a panel scale. Metallization has been done with a sputtered Cu plating base followed by Cu plating. A conductor line and LGA pads have been structured by etching using a dry film photoresist and, again, laser direct imaging. As a last step before the package singulation, a solder mask is applied and structured on the wafer backside.

## 4. Results and Discussion

For the package evaluation and quality inspection, the finalized panels and singulated packages have been analyzed non-destructively and destructively. In Figure 7, an example of a fully processed 457 × 305 mm^2^ panel is shown. The panels could be successfully processed without, for instance, cracking the panel.

In Figure 8, detailed pictures of a daisy chain and a functional package are depicted. A good alignment of the die pads and SMD terminals are visible as a result of the adaptive routing of the RDL to the real die positions, as described above.

The packages have been also characterized by electrical testing. For the daisy chain packages, the resistance of the chip interconnections as well as the capacity of the embedded capacitors have been measured. In Figure 9, the measurement results of 350 packages are shown. For the resistance, we detected a mean value of 0.93 Ohm with a standard deviation of 0.18 Ohm, and for the capacitor we detected a mean value of 46.3 nF with a standard deviation of 0.48 nF. A high package yield could be achieved, and the electrical measurements show good results, while still indicating some potential for optimization in the RDL processing. 

An X-ray computer tomography (CT) was used for a non-destructive analysis of the overall package integrity with a focus on the metal structures and interconnects. In Figure 10, the results of a daisy chain package are depicted. A 3D view, along with virtual cross sections, proof the well aligned RDL structures as well as the integrity of the via connection to the die pads and both the SMD terminals and conductor lines.

In addition, metallographic cross sections were prepared to investigate the overall package quality. A cross section through the die and SMD component is depicted in Figure 11. No delaminations can be observed between the die and epoxy molding compound and the dielectric layer of the RDL, indicating a good adhesion between all of the materials and layers. A good alignment and well-formed via interconnection on the die pads is also visible. 

In summary, a good package quality was achieved with the FOWLP approach, and the process could be successfully transferred from the wafer formats to the panel formats.

## 5. Conclusions

Fan-Out Wafer and Panel Level Packaging is one of the latest trends in microelectronics packaging with the potential for miniaturization but also for heterogeneous packaging. In this study, the “mold first” approach was chosen for an ASIC package development with integrated SMD capacitors. The process developments and the successful overall proof of concept for the fan-out wafer/panel level packaging approach were conducted on a 200 mm wafer size. In a second step, the technology was scaled up to a 457 × 305 mm^2^ panel size using the same materials, equipment and process flow, demonstrating the low cost and large area capabilities of the approach. Hence, this technology is well suited for applications such as the targeted energy harvester, where heterogeneous components have to be integrated into a miniaturized system.

## Figures and Tables

**Figure 1 micromachines-10-00342-f001:**
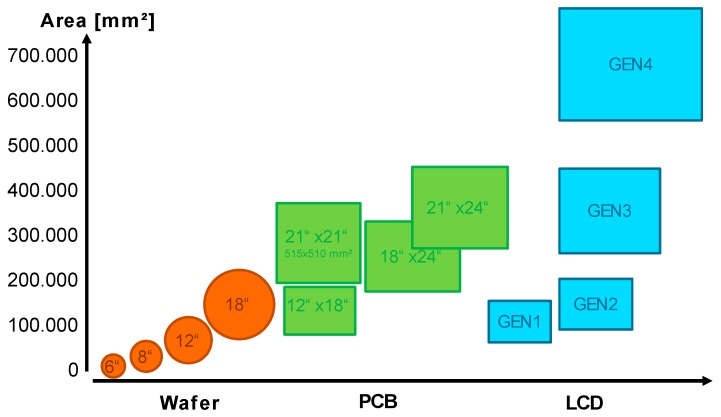
Existing wafer and panel sizes influencing fan-out panel level packaging developments.

**Figure 2 micromachines-10-00342-f002:**
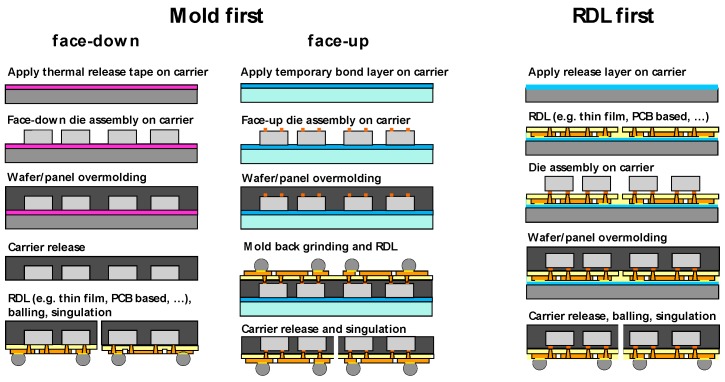
Fan-out wafer/panel level packaging process flow options.

**Figure 3 micromachines-10-00342-f003:**
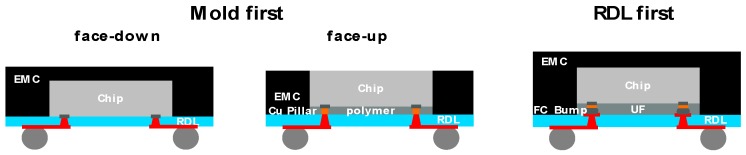
Fan-out wafer/panel level package structures from different process flow options.

**Figure 4 micromachines-10-00342-f004:**
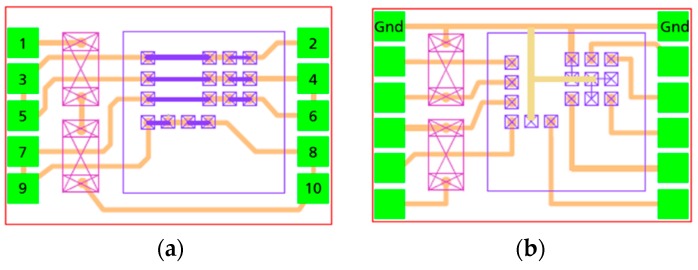
Package layout; (**a**) daisy chain package, (**b**) functional.

**Figure 5 micromachines-10-00342-f005:**
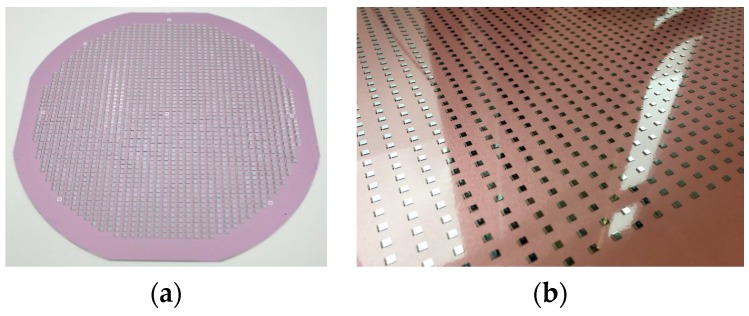
Die and SMD assembly on the carrier; (**a**) 200 mm wafer, (**b**) detail of the 457 × 305 mm^2^ panel.

**Figure 6 micromachines-10-00342-f006:**
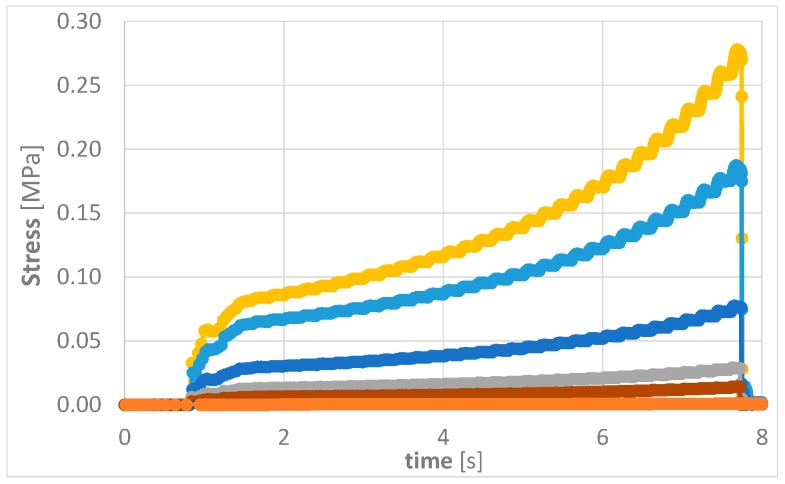
Existing wafer and panel sizes influencing fan-out panel level packaging developments.

**Figure 7 micromachines-10-00342-f007:**
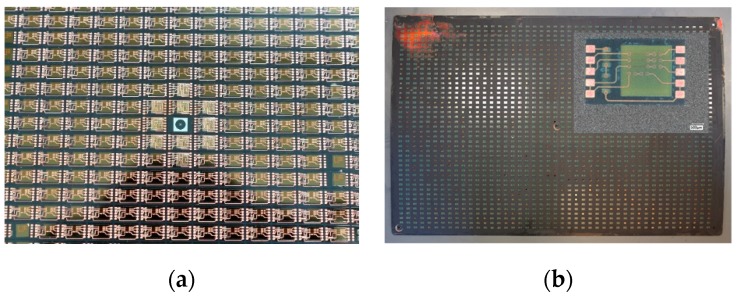
Processed fan-out panel; (**a**) panel detail, (**b**) panel overview with package detail (ASIC with SMD components).

**Figure 8 micromachines-10-00342-f008:**
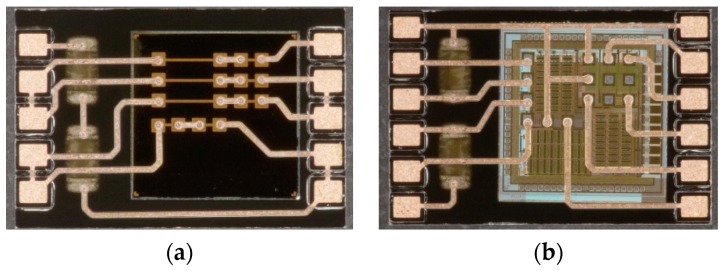
Manufactured ASIC packages by FOWLP/PLP; (**a**) daisy chain package, (**b**) functional package.

**Figure 9 micromachines-10-00342-f009:**
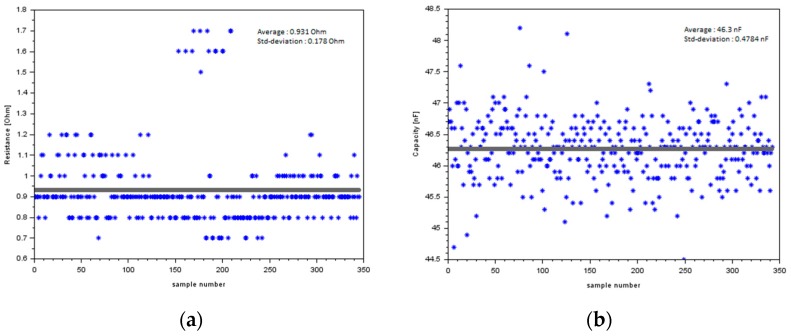
Electrical test results of the daisy chain packages; (**a**) daisy chain resistance of chip to package interconnects, (**b**) capacity of integrated capacitors.

**Figure 10 micromachines-10-00342-f010:**
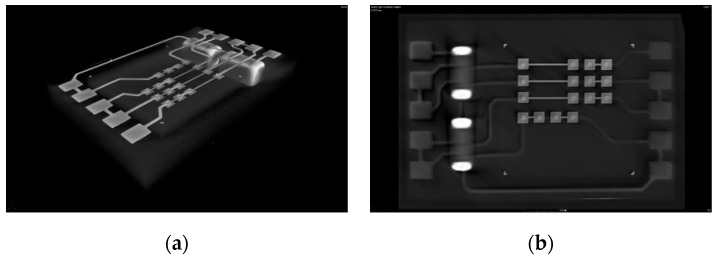
X-ray CT package analysis; (**a**) package overview, (**b**) package top layer; (**c**) virtual cross section through chip interconnects, (**d**) virtual cross section through SMD interconnects.

**Figure 11 micromachines-10-00342-f011:**
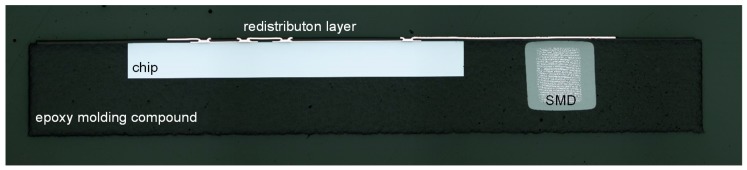
Package cross section; ASIC with SMD component embedded in the molding compound and connected by a thin film redistribution layer.

**Table 1 micromachines-10-00342-t001:** Material data for epoxy molding compound used for encapsulation (data sheet values).

Type	Liquid
Filler content [wt%]	88
Maximum filler size [µm]	25
Cure condition	In Mold Cure: 125 °C/10 min Post Mold Cure: 150 °C/60 min
Specific gravity @ 25 °C	1.96
Tg (DMA) [°C]	150
CTE_1_ [ppm/K]	8
CTE_2_ [ppm/K]	41

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
