# Peer review of "Fan-Out Wafer and Panel Level Packaging as Packaging Platform for Heterogeneous Integration"

_micromachines, 2019, doi:10.3390/mi10050342_

Round 1
Reviewer 1 Report
Report on paper "Fan-out Wafer and Panel Level Packaging for Miniaturized Energy Harvester Packaging" submitted by Braun et al., for publication in MDPI Micromachines (micromachines- 479552).
The authors validated successfully the proof of concept for the LED Fan-out Wafer/Panel Level Packaging approach, which has been done on 200 mm wafer size. The proposed technology can be suitable for applications such as miniaturized energy harvesters. The paper is interesting and globally well organized. Nevertheless, several points must be improved and the authors should address the following comments:
1. The focus of the paper is packaging and not energy harvesting. In my opinion, the word “energy harvester” should be deleted from the title and from the keywords. This can be clearly justified by the fact that energy harvesting is a possible application among many other applications.
2. The abstract must be modified and should contain the highlights of the paper.
3. In the introduction, the literature survey lacks of references in the field of MEMS packaging, which is a topic deeply investigated in the recent past.
4. The originality of this paper must be highlighted in the end of the introduction and with respect to the recent published papers of the authors.
5. The captions of the figures should contain full description of all the important details.
Author Response
Report on paper "Fan-out Wafer and Panel Level Packaging for Miniaturized Energy Harvester Packaging" submitted by Braun et al., for publication in MDPI Micromachines (micromachines- 479552).
The authors validated successfully the proof of concept for the LED Fan-out Wafer/Panel Level Packaging approach, which has been done on 200 mm wafer size. The proposed technology can be suitable for applications such as miniaturized energy harvesters. The paper is interesting and globally well organized. Nevertheless, several points must be improved and the authors should address the following comments:
1. The focus of the paper is packaging and not energy harvesting. In my opinion, the word “energy harvester” should be deleted from the title and from the keywords. This can be clearly justified by the fact that energy harvesting is a possible application among many other applications.
-> Title has been changed to “Fan-out Wafer and Panel Level Packaging as Packaging Platform for Heterogeneous Integration”
2. The abstract must be modified and should contain the highlights of the paper.
-> A short paragraph has been added on the results and highlights.
3. In the introduction, the literature survey lacks of references in the field of MEMS packaging, which is a topic deeply investigated in the recent past.
-> Some more references has been added, but not too many regarding MEMS packaging as the focus of the paper was not the packaging of the MEMS.
4. The originality of this paper must be highlighted in the end of the introduction and with respect to the recent published papers of the authors.
-> Slightly modified and some references are given.
5. The captions of the figures should contain full description of all the important details.
-> modified
Reviewer 2 Report
This paper presents FOWLP considered for packaging MEMS energy harvesters. At this point in time, any research into wafer level packaging for MEMS energy harvesting would be of great interest to the community as this is still an under-researched area. FOWLP itself is not new and has seen a growing adoption in the wider semiconductor IC fields, however, it is indeed a novel attempt to implement for energy harvester. The advantage is obviously smaller packaged size. However, I do not feel the authors have sufficiently addressed the challenges that are unique to energy harvesters. The figures shown also don't clearly show or indicate an operational energy harvester? Please kindly update this to show how the harvesters operate within the package.
The main challenge is that harvesters are not passive IC components, they are moving mechanical components that can have a much larger travel than other MEMS oscillators such as accelerometers or gyroscopes. MEMS harvester oscillators can have travel of 10's um to even 100's um. This needs to be accommodated within the package, which at the moment it is not clear how this is considered? This is not something that can be considered separately, but needs to be holistically designed together with package cavity.
Also, does the present work control the air pressure within the package? With confined cavity, squeeze film damping is a big issue long known within MEMS oscillators. Therefore, this is another important consideration that needs to be holistically considered.
This is a very interesting work, and I would advise the authors to update and consider the above points. At the present, these are not clearly addressed in the paper.
Author Response
This paper is a part of a Special Issue with focus on a miniaturized energy harvester system which is the summary of a European funded project. More Information on the harvester itself can be found in one or two other papers of this Special Issue. Due to the overall results of the MEMS harvester this approach was not followed until the end of the project including packaging and integration. Therefore this paper can not present these results.
However, MEMS packaging is an important but also challenging topic. We are working in different projects on that and results will be published. But these results can not be included in this paper.

Round 2
Reviewer 1 Report
The authors have addressed my comments sufficiently to recommend publication of the paper in its current form.
Reviewer 2 Report
I am satisfied with authors' response and I recommend for publication.